# Myc controls NK cell development, IL-15-driven expansion, and translational machinery

Hanif J Khameneh[1,*] , Nicolas Fonta[1,*], Alessandro Zenobi[1] , Charlène Niogret[2], Pedro Ventura[1], Concetta Guerra[1] , Ivo Kwee[3], Andrea Rinaldi[4], Matteo Pecoraro[1], Roger Geiger[1,4] , Andrea Cavalli[1,5], Francesco Bertoni[4,6], Eric Vivier[7,8,9], Andreas Trumpp[10,11], Greta Guarda[1]

**MYC is a pleiotropic transcription factor involved in cancer, cell proliferation, and metabolism. Its regulation and function in NK cells, which are innate cytotoxic lymphocytes important to control viral infections and cancer, remain poorly defined. Here, we show that mice deficient for Myc in NK cells presented a severe reduction in these lymphocytes. Myc was required for NK cell development and expansion in response to the key cytokine IL-15, which induced Myc through transcriptional and posttranslational mechanisms. Mechanistically, Myc ablation in vivo largely impacted NK cells' ribosomagenesis, reducing their translation and expansion capacities. Similar results were obtained by inhibiting MYC in human NK cells. Impairing translation by pharmacological intervention phenocopied the consequences of deleting or blocking MYC in vitro. Notably, mice lacking Myc in NK cells exhibited defective anticancer immunity, which reflected their decreased numbers of mature NK cells exerting suboptimal cytotoxic functions. These results indicate that MYC is a central node in NK cells, connecting IL-15 to translational fitness, expansion, and anticancer immunity.**

## Introduction

NK cells, which are important to control selected cancers and infections, express multiple germ-line-encoded activating and inhibitory receptors. The latter are engaged by major histocompatibility complex class I (MHCI) molecules, whose expression can be lost by hazardous cells trying to evade T cell responses. Conversely, activating receptors are engaged by ligands expressed by stressed cells, including infected

and transformed ones. NK cell cytotoxic activity therefore depends on the imbalance towards activating signals (Narni-Mancinelli et al, 2011). Recently, the generation of chimeric antigen receptor-engineered NK cells and of "multifunctional engagers" promoting NK cell activation against tumoral cells further fueled the interest in these lymphocytes (Gardiner, 2019).

NK cell development takes place in the BM and it is largely driven by IL-15 (Kennedy et al, 2000). In the mouse, CD27 single-positive (CD27SP) cells are immature and feature high proliferative and metabolic activity. NK cells further mature through a CD27 and CD11b double-positive (DP) stage to a terminally differentiated CD11b single-positive (CD11bSP) subset, which represents the largest peripheral population and is characterized—unless stimulated—by a quiescent state (Marcais et al, 2014). This is important, as these lymphocytes might otherwise trigger tissue damage or give rise to malignancies (Gianchecchi et al, 2021). These include NK cell-derived lymphoma, in which increased levels of the proto-oncogene MYC correlate with poor prognosis (Huang et al, 2014; Wang et al, 2017; Xiong et al, 2020).

MYC is a well-known transcription factor involved in various types of cancers (Dhanasekaran et al, 2022). Even if MYC-binding sites are common throughout the genome, the landscape of its target genes varies based on the investigated cell type (Campbell & White, 2014; Link & Hurlin, 2015). Although the best characterized process controlled by MYC is proliferation, many others are regulated by this transcription factor (Dang et al, 2006; Bretones et al, 2015). In NKT and T cells, its role in development, activation, and proliferation (Bianchi et al, 2006; Dose et al, 2006, 2009; Jiang et al, 2010; Nozais et al, 2021) and in orchestrating specific metabolic configurations (Wang et al, 2011; Marchingo et al, 2020; Saravia et al, 2020) has emerged in the past years.

As for NK cells, exposure to inflammatory cytokines IL-12 and IL-2 or IL-18 has been shown to induce MYC, which in turn increased

[1]Università della Svizzera italiana, Faculty of Biomedical Sciences, Institute for Research in Biomedicine, Bellinzona, Switzerland  [2]Department of Biochemistry, University of Lausanne, Epalinges, Switzerland  [3]BigOmics Analytics SA, Lugano, Switzerland  [4]Università della Svizzera italiana, Faculty of Biomedical Sciences, Institute of Oncology Research, Bellinzona, Switzerland  [5]Swiss Institute of Bioinformatics, Lausanne, Switzerland  [6]Oncology Institute of Southern Switzerland, Ente Ospedaliero Cantonale, Bellinzona, Switzerland  [7]Aix-Marseille Université, Centre National de la Recherche Scientifique, Institut National de la Santé et de la Recherche Médicale, Centre d'Immunologie de Marseille-Luminy, Marseille, France  [8]Innate Pharma Research Laboratories, Marseille, France  [9]APHM, Hôpital de la Timone, Marseille-Immunopôle, Marseille, France  [10]Division of Stem Cells and Cancer, DKFZ, Heidelberg, Germany  [11]HI-STEM: The Heidelberg Institute for Stem Cell Technology and Experimental Medicine gGmbH, Heidelberg, Germany

Correspondence: greta.guarda@irb.usi.ch
*Hanif J Khameneh and Nicolas Fonta contributed equally to this work

● *Ncr1cre Myc*wt/wt   ● *Ncr1cre Myc*fl/fl

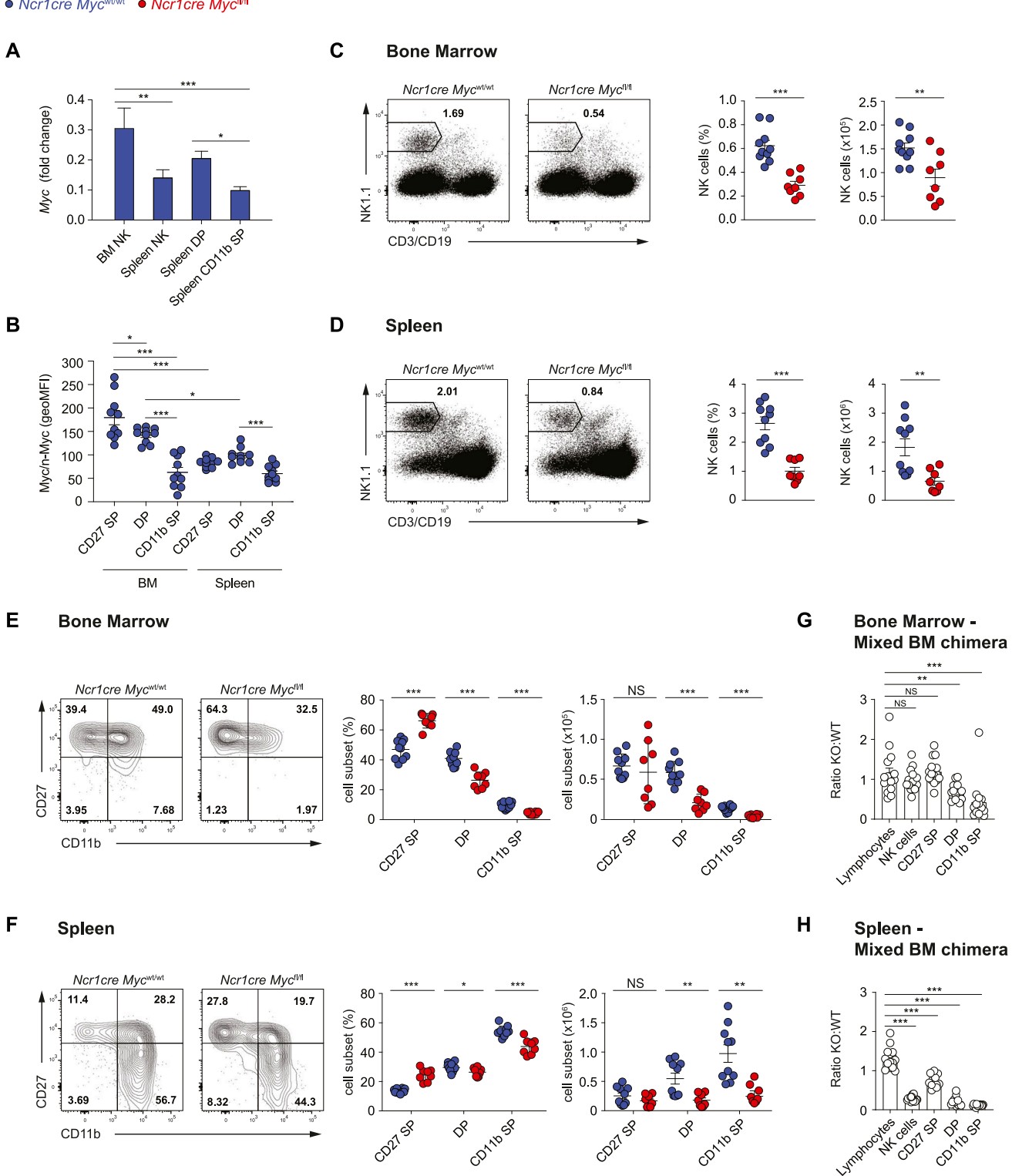

**Figure 1.   Ncr1cre Myc**fl/fl **mice present severely decreased NK cell numbers.**
**(A)** qRT–PCR analysis (normalized to *Polr2a* and *18s*) is shown for *Myc* mRNA in sorted BM NK cells (NK1.1+Ncr1+CD3/19−), and splenic total (NK1.1+Ncr1+CD3/19−), double-positive (DP; CD27+CD11b+), and CD11b single-positive (CD11b SP; CD27−CD11b+) NK cells from C57BL/6 mice. **(B)** Myc expression levels in the CD27 single-positive (CD27 SP; CD27+CD11b−), DP, and CD11b SP NK cell subsets (NK1.1+CD3−) from the BM and spleen are expressed as geometric MFI. **(C, D)** A representative flow cytometry plot and a quantification of percentages and numbers of NK cells in the BM ((C), CD122+NK1.1+Ncr1+CD3/19−) and spleen ((D), NK1.1+Ncr1+CD3/19−) of the indicated mice are shown. **(E, F)** A representative flow cytometry plot and a quantification of percentages and numbers of BM (E) and splenic (F) CD27 SP, DP, and CD11b SP NK cell populations. **(G, H)** Graphs depict the ratio of *Ncr1cre Myc*fl/fl over *Ncr1cre Myc*wt/wt for lymphocytes, total, CD27 SP, DP, and CD11b SP NK cells (CD122+NK1.1+CD3/19−) in the BM (G) and

glycolysis, mitochondrial activity, and effector functions (Loftus et al, 2018; Dong et al, 2019). Stimulation with IL-15, which is crucial for both NK cell homeostasis and effector function, was shown to increase *MYC* transcription (Cichocki et al, 2009; Ma et al, 2022), which was reported to mediate the up-regulation of NK cell inhibitory receptors (Cichocki et al, 2009) and maintain normal numbers of peripheral NK cells (Dong et al, 2019). Whereas these notions suggest that MYC represents a key factor in NK cells and in mediating the effects of IL-15, we still lack information on its regulation, transcriptional targets, and specific function in these lymphocytes.

We thus set out to investigate MYC expression and the transcriptional landscape regulated by this transcription factor in NK cells. We found that Myc is highly expressed in the more immature BM subsets, whereas its levels are significantly lower in splenic NK cells. In line with this, IL-15 induced and stabilized MYC through distinct mechanisms in both murine and human NK cells. Mice lacking Myc in NK cells presented a substantial reduction of NK cell numbers in the periphery, which was due to a cell-intrinsic defect in developing NK cells. Mechanistically, Myc-deficient NK cells presented a strongly perturbed network of genes coding for ribosome and translation-related proteins, which substantially affected their translation rate and expansion upon IL-15 stimulation, a phenomenon largely recapitulated in human NK cells upon MYC blockade. Together, our results indicate a key role for Myc in regulating translation capacity during IL-15-induced proliferation and the tightly coupled developmental program in NK cells, eventually leading to defective cell numbers and anticancer immunity.

## Results

### Myc is required for mature NK cell development

We assessed the expression of *Myc* mRNA by quantitative real-time RT–PCR (qRT–PCR) in BM and splenic NK cells, and splenic NK cells of the DP and CD11b SP subsets (Fig 1A). Although *Myc* expression was detected in all tested subpopulations, the highest transcript levels were found in BM NK cells. We therefore tested the expression of Myc at the protein level and found highest Myc expression in CD27SP and DP in the BM, whereas BM-derived CD11b SP and splenic NK cells showed lower levels (Figs 1B and S1A). In line with analyses in human NK cells (Collins et al, 2019), these results suggest that Myc is important for NK cell with high proliferative potential, whereas its reduced levels promote NK cell quiescence in the periphery.

We thus generated *Ncr1cre Myc*<sup>fl/fl</sup> mice, in which, Myc expression is deleted in NK cells at the immature stage (Trumpp et al, 2001; Narni-Mancinelli et al, 2011). Deletion was corroborated by qRT–PCR

(Fig S1B). Given that the antibody used for Myc detection can recognize both Myc and n-Myc, generation of Myc-deficient NK cells also allowed us to ensure specificity of the staining for Myc in these lymphocytes (Fig S1C). Our analyses of *Ncr1cre Myc*<sup>fl/fl</sup> mice further showed that the percentage and number of NK cells were significantly reduced as compared with control mice, both in the BM and in the spleen (Figs 1C and D and S1D; a typical gating strategy for NK cells in the BM and spleen is shown in Fig S1E). Whereas the effect of Myc deficiency on the numbers of CD27 SP NK cells was variable, which is in line with the recent activation of Cre expression in this subset, a prominent decrease in NK cell numbers at the more mature CD11b<sup>+</sup> maturation stages was observed in the BM and spleen (Figs 1E and F and S1D). Mixed BM chimeras demonstrated that deletion of this transcription factor affected the numbers of CD11b<sup>+</sup> NK cells in a cell-intrinsic manner (Fig 1G and H). Interestingly, liver conventional NK cells were also markedly decreased in percentage and number, whereas the CD49a<sup>+</sup> type 1 innate lymphoid cells (ILC1s), in which *Ncr1cre*-mediated deletion is also expected (Ali et al, 2016; Cuff & Male, 2017; Ducimetiere et al, 2021), did not show a similar decrease (Fig S1F). Together, these results indicate that Myc is critical for the development of conventional NK cells.

### Myc is integral to NK cell proliferation

We thus asked whether the developmental defect of Myc-deficient NK cells was related to a defect in proliferation. We found that the highly dividing CD27 SP NK cells and the DP NK cells from *Ncr1cre Myc*<sup>fl/fl</sup> mice presented strongly reduced levels of the proliferation marker Ki67 as compared with the control counterparts in the BM and spleen (Fig 2A and B). Of note, the levels of the antiapoptotic protein Bcl2 were not decreased in these cells (Fig 2C). Furthermore, we confirmed that reduced percentages of Ki67-positive cells among CD27 SP and DP Myc-deficient NK cells were observed in mixed BM chimeras (Fig 2D and E). This indicates that, in particular, CD27SP NK cells rely on Myc to expand, explaining the decreased numbers of Myc-deficient peripheral NK cells.

Given the importance of IL-15 in driving NK cell development in vivo (Lee et al, 2011; Marcais et al, 2013), we assessed in vitro the response of Myc-deficient NK cells to this cytokine. At an IL-15 concentration not inducing cell division, the survival of these cells was comparable with the one of control NK cells (Fig 2F). Moreover, in response to IL-15 doses inducing cell division, the percentage of live cells was comparable between the two genotypes, whereas Myc-deficient NK cells exhibited severely compromised expansion, hinting to an important proliferative defect (Fig 2F). Of note, a similar result was obtained also in response to IL-2, a cytokine sharing two receptor subunits with IL-15 (Fig 2F).

spleen (H) of mixed BM chimeras. **(A, B, C, D, E, F, G, H)** Results depict mean ± SD (n = 3 technical replicates) and are representative of at least two experiments (A), mean ± SEM of n = 10 (B), n = 8–10 (C, D, E, F), and n = 15 (G, H) mice per condition and are a pool of two experiments (B, C, D, E, F, G, H). **(B, C, D, E, F, G, H)** Each symbol represents an individual mouse. **(A, B, C, D, E, F, G, H)** Statistical comparisons are shown; *$P \leq 0.05$, **$P \leq 0.01$, ***$P \leq 0.001$, and NS, non-significant; one-way ANOVA test (A, B), t test, unpaired (C, D, E, F, G, H). **(A, B)** Only statistically significant differences are shown in (A) and only statistically significant differences comparing NK cell subsets in the same organ and the same subsets across the two organs are shown (B).

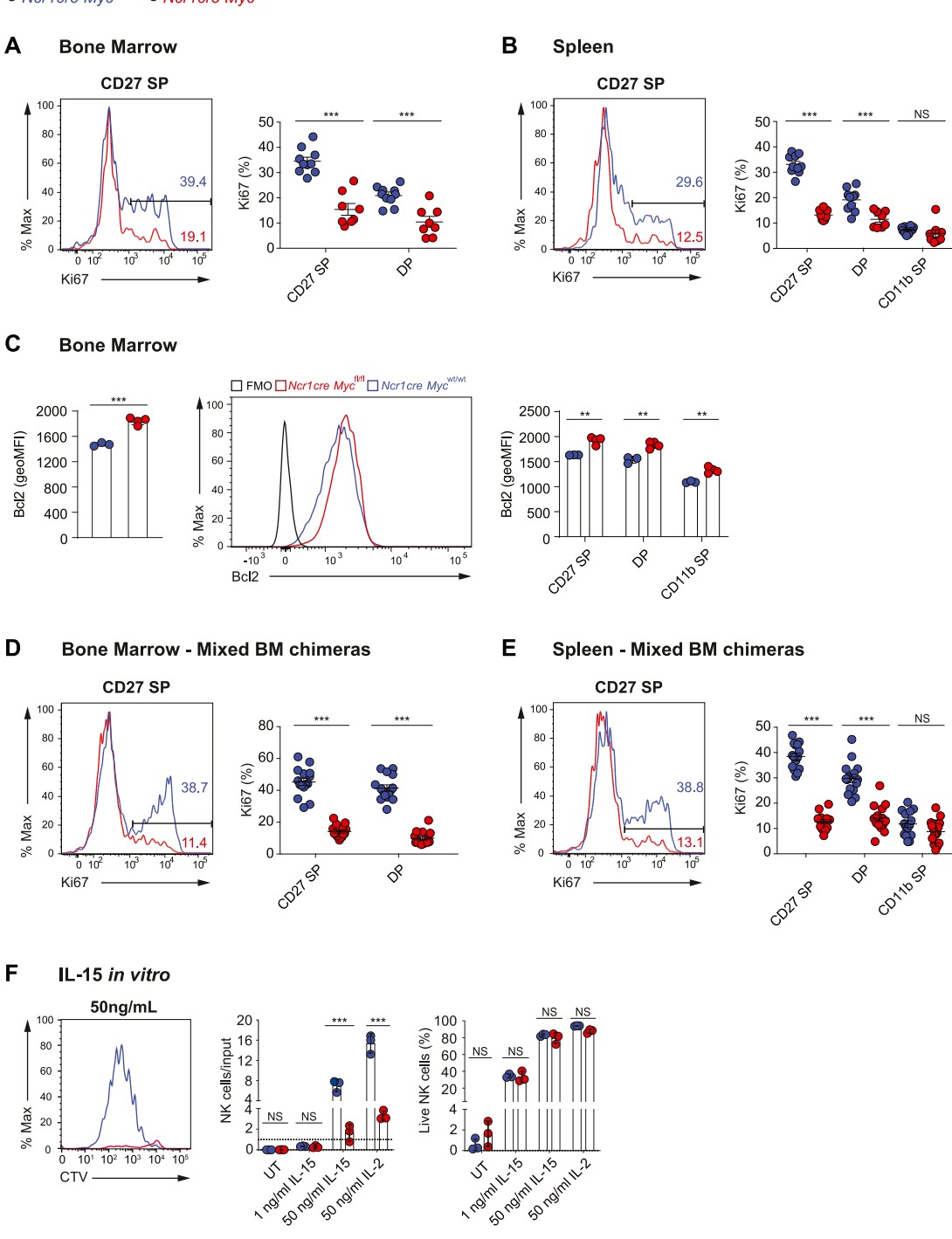

**Figure 2. Myc-deficient NK cells exhibit a defective response to IL-15.**

**(A, B)** Percentages of Ki67⁺ cells among the indicated subsets are shown in the BM (A) and spleen (B) of *Ncr1cre Myc*^fl/fl^ and *Ncr1cre Myc*^wt/wt^ mice. **(C)** Graphs show geometric mean and a representative flow cytometry plot of Bcl2 expression in BM NK cells (NK1.1⁺Ncr1⁺CD3⁻) and in CD27 SP, DP, and CD11b SP NK cells of indicated genotypes. **(D, E)** Percentages of Ki67⁺ cells among the indicated subsets in the BM (D) and spleen (E) of *Ncr1cre Myc*^fl/fl^ and *Ncr1cre Myc*^wt/wt^ mixed BM chimeras. **(F)** Splenocytes from *Ncr1cre Myc*^fl/fl^ and control mice were cultured in vitro in the presence of the indicated doses of IL-15 or IL-2. Graphs show a representative cytometric profile of cell trace violet dilution by NK cells (gated as NK1.1+ CD3/19⁻), the ratio of live NK cell numbers (PI negative) normalized to the numbers of seeded NK cells of the same genotype, and the percentage of live cells among the NK population of the indicated genotypes after 4 d of culture. **(A, B, C, D, E, F)** Results represent mean ± SEM of n = 8–10 mice per genotype (A, B), ± SEM of n = 3–4 mice per genotype (C), n = 15 chimeric mice (D, E), mean ± SD of n = 3 technical replicates and are a pool of two independent experiments (A, B, D, E) or representative of at least two independent experiments (C, F) and each symbol represents an individual mouse (A, B, C, D, E). **(A, B, C, D, E, F)** Statistical comparisons are shown; **P ≤ 0.01, ***P ≤ 0.001, and NS, non-significant; *t* test, unpaired.

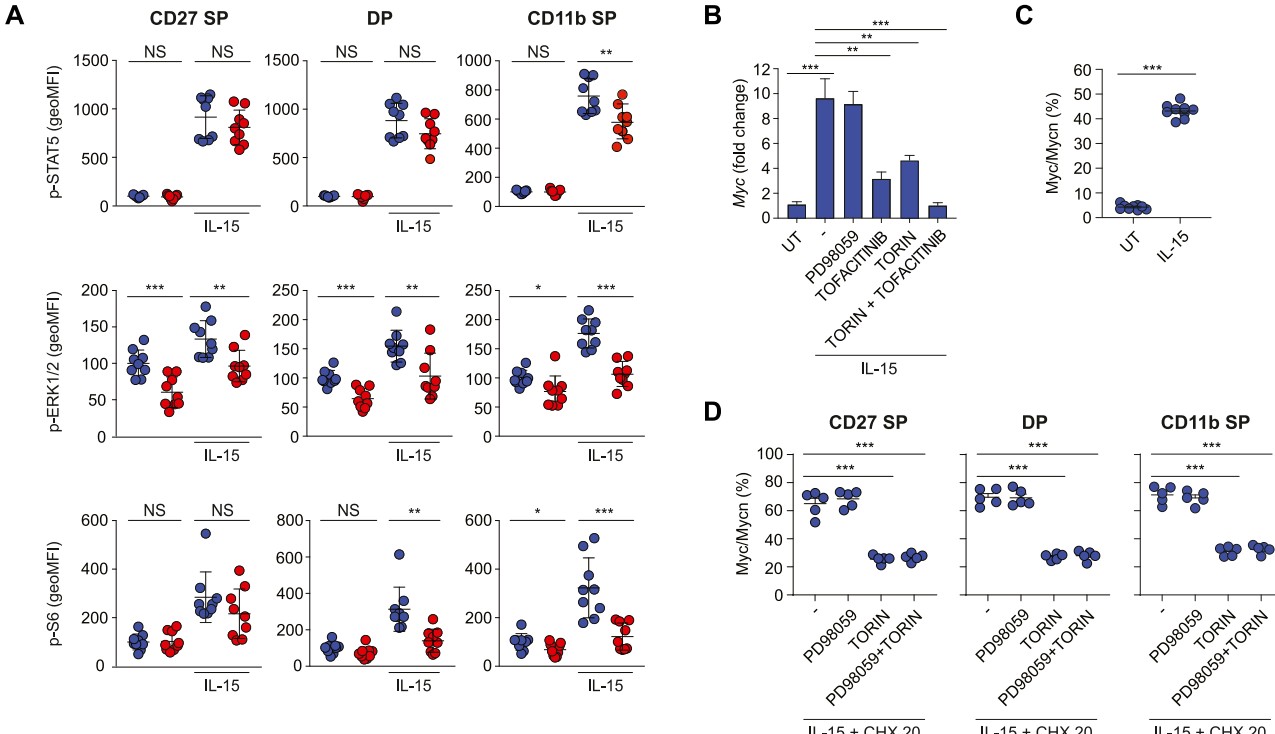

**Figure 3. Myc protein levels are induced by IL-15 through mTOR and ERK.**
**(A)** Splenocytes from *Ncr1cre Myc*<sup>fl/fl</sup> and *Ncr1cre Myc*<sup>wt/wt</sup> mice were cultured in vitro for 40 min in the presence of 50 ng/ml IL-15 or left untreated. Phosphorylation of STAT5, ERK, and S6 were measured by flow cytometry in CD27 SP, DP, and CD11b SP NK cell subsets (CD122$^+$, NK1.1$^+$, and CD3/19$^-$; the average of the unstimulated *Ncr1cre Myc*<sup>wt/wt</sup> was set as 100%). **(B)** qRT–PCR analysis (normalized to *Polr2a/18 s*) is shown for *Myc* mRNA in purified WT NK cells pretreated or not with Torin2 (250 nM), PD98059 (10 μM), tofacitinib (90 nM) or combinations thereof, and stimulated with 50 ng/ml of IL-15 for 2 h. **(C)** Splenocytes from control mice were stimulated with 50 ng/ml of IL-15 or left untreated for 2 h and the percentage of Myc/n-Myc-positive cells NK cells (NK1.1$^+$CD3$^-$) is shown. **(D)** Splenocytes from control mice were stimulated with 50 ng/ml of IL-15 for 2 h and then treated with cycloheximide (20 μg/ml), Torin2 (250 nM), and/or PD98059 (10 μM), as indicated. Graphs show the percentage reduction in Myc after 70 min (measured as geometric MFI) in the indicated NK cell subsets (NK1.1$^+$). **(A, B, C, D)** Results depict mean ± SEM of n = 9 mice per genotype and are a pool of two experiments (A), mean ± SD of n = 3 technical replicates and are representative of at least two experiments (B), mean ± SD of n = 9 mice and are a pool of two independent experiments (C), mean ± SEM of n = 5 mice and are representative of two independent experiments (D). **(A, C, D)** Each symbol represents an individual mouse. **(A, B, C, D)** Statistical comparisons are shown; *$P$ ≤ 0.05, **$P$ ≤ 0.01, ***$P$ ≤ 0.001, and NS, non-significant; *t* test, unpaired (A, B) or paired (C, D) and only statistically significant differences are shown (B, D).

## Myc is regulated by multiple IL-15-triggered signaling pathways

Having observed that Myc-deficient NK cells exhibit an expansion defect in response to IL-15 and IL-2 stimulations, we measured the expression of the two shared receptor chains CD122 (IL2Rβ/IL15Rβ) and CD132 (common γ chain). CD122 showed a moderately reduced expression on Myc-deficient NK cells, whereas we did not detect differences in CD132 levels (Fig S2A and B). Next, we thought to assess the phosphorylation of signal transducer and activator of transcription 5 (STAT5), extracellular signal-regulated kinase (ERK), and S6, key signaling pathways induced by IL-15 in these cells. Multiple cascades were affected in the more mature Myc-deficient NK cell subsets (Fig 3A). Being the earliest deleted subpopulation and the one with the strongest proliferation defect (Fig 2A, B, D, and E), we were however particularly interested in the CD27 SP subset, which only presented a reduction in ERK signaling noticeable already ex vivo (Fig 3A). Immunoblot analysis of splenic Myc-deficient

NK cells did not reveal major differences in the expression of total STAT5, S6 or ERK2 (Fig S2C).

As previous results showed that an even stronger ERK signaling defect in Shp-2-deficient NK cells was not associated to an impaired development (Niogret et al, 2019), we decided to investigate if instead IL-15 was regulating Myc. In line with previous findings, we observed that IL-15 stimulation markedly increased *Myc* transcript levels (Fig 3B) (Cichocki et al, 2009; Mishra et al, 2012; Gotthardt et al, 2016; Pinz et al, 2016; Villarino et al, 2017). The inhibitors tofacitinib (pan JAK inhibitor), PD98059 (mitogen-activated protein kinase kinase inhibitor), and Torin2 were used to block the JAK, ERK, and mammalian target of rapamycin (mTOR) pathways, respectively (Fig S2D) (Pende et al, 2004; Meyer & Levine, 2014). Whereas tofacitinib and Torin2 affected *Myc* mRNA induction, concomitant treatment almost prevented it (Fig 3B). Stimulation of splenic NK cells also increased Myc protein levels (Fig 3C). To distinguish the contribution of the posttranslational regulation of Myc from its transcriptional

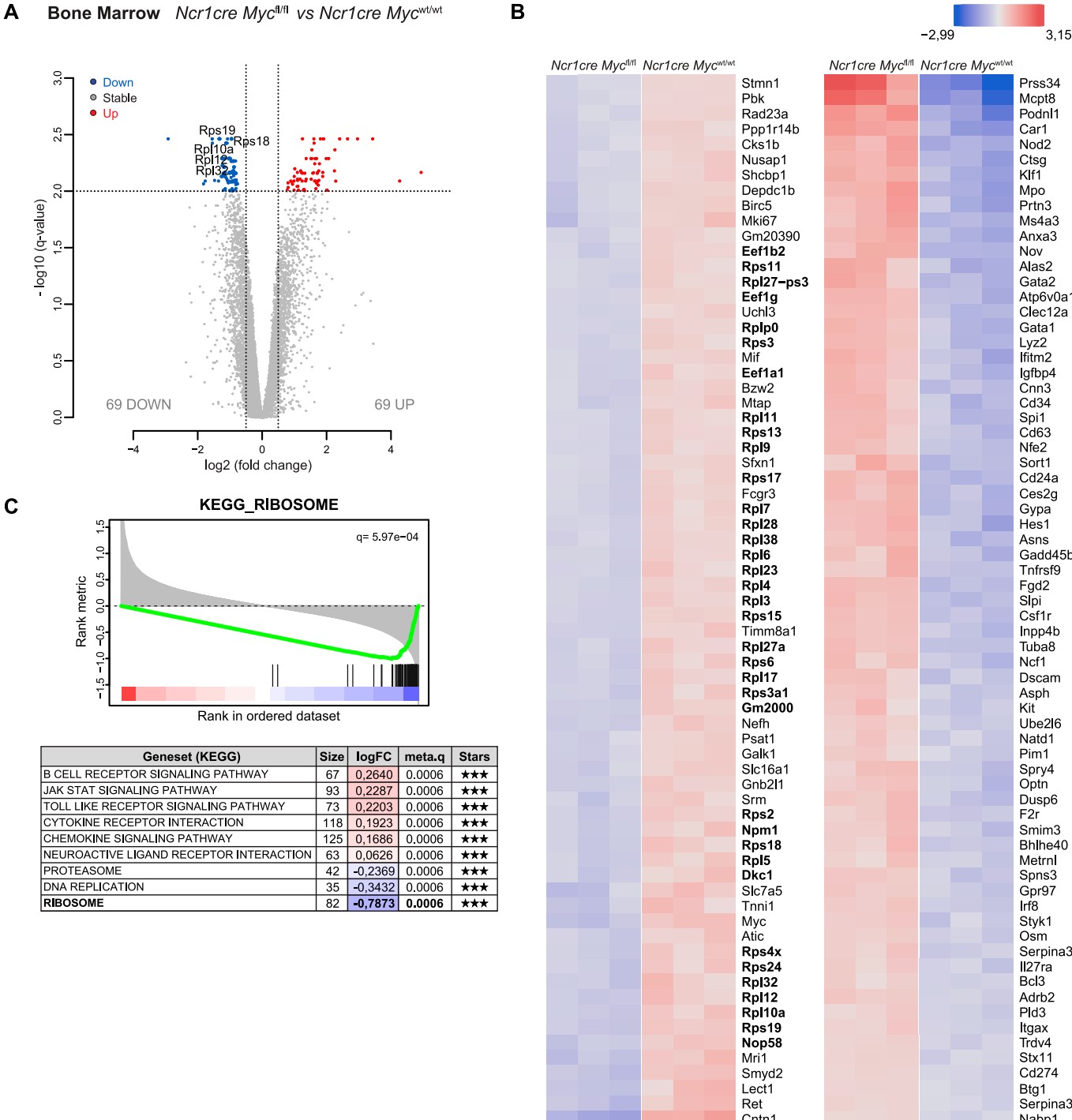

**Figure 4. Myc-deficient NK cells present defective expression of genes related to translation.**
**(A, B, C)** RNA sequencing analyses of CD27SP NK cells (sorted as NK1.1⁺CD27⁺CD11b⁻CD3/19⁻) harvested from the BM of *Ncr1cre Myc*^fl/fl and *Ncr1cre Myc*^wt/wt mice.
**(A)** Volcano plot displaying differential expressed genes; the vertical axis (y-axis) displays the significance as the $\log_{10}$ (P-value), and the horizontal axis (x-axis) displays effect size as the $\log_2$ fold change value (FDR = 0.01; logFC threshold = 0.5; genes shown are significant for the three following statistical methods; DESeq2 [Wald test], edgeR, limma-trend). **(B)** Hierarchical clustering of the 138 significantly differentially expressed genes. **(C)** The table shows the top significant KEGG gene sets (q ≤ 0.0006 with fGSEA, fisher, and GSVA methods—indicated under Stars), their fold-change, and the number of genes in each. Enrichment plot for ribosome KEGG gene set is shown.

induction, we treated IL-15-stimulated NK cells with cycloheximide to prevent further protein synthesis and studied the effects of mTOR and ERK, two pathways implicated in Myc stabilization (Sears et al, 2000; Gustafson & Weiss, 2010), on Myc levels (Fig 3D). We observed that Myc degradation was accelerated by the blockade of mTOR, whereas ERK pathway inhibition exerted negligible effects. These results indicate that, in IL-15-treated NK cells, JAK-STAT and mTOR are mostly responsible for *Myc* induction, whereas mTOR for Myc stability.

### Myc regulates genes related to translation

To delineate the role of Myc in NK cell expansion and development, we performed RNA sequencing (RNA-seq) analyses. Being the earliest deleted subset and presenting the strongest proliferation defect (Fig 2A, B, D, and E), we selected Myc-deficient and -sufficient CD27SP NK cells harvested from the BM. Paired differential expression analyses identified 69 genes significantly down-regulated and 69 genes significantly up-regulated in the absence of Myc (Fig 4A). We next performed a hierarchical clustering of these genes (Fig 4B). Notably, 36 of the down-regulated genes coded for ribosomal subunits, translation elongation factors, and proteins involved in ribosomal formation (Fig 4B; indicated in bold). Among the pathways revealed by Gene set enrichment analysis with highest significance (Kanehisa & Goto, 2000), the most deregulated referred to "Ribosome" (Fig 4C). Of note, this finding is in line with work by Tang et al (2021), who demonstrated a defect in translation-related genes in Myc-deficient bulk splenic NK cells. In keeping with translation-related genes, we also identified *Slc7a5*, a Myc target involved in amino acid transport, among the down-regulated genes in Myc-deficient immature NK cells (Fig 4B) (Marchingo et al, 2020). In line with our flow cytometric analyses, we observed down-regulated expression of the gene coding for Ki67 (*Mki67*), which has also been linked to nucleologenesis (Fig 4B) (Schmidt et al, 2003; Rahmanzadeh et al, 2007). Instead, among these significantly altered genes, we did not observe canonical Myc-responsive genes involved in cell cycle. We therefore specifically looked for their expression, revealing that part of them was tendentially—but not significantly—altered, whereas others were not (Fig S3A) (Robson et al, 2011; Mishra et al, 2012; Bretones et al, 2015). Similar results were observed for other pathways classically associated to Myc, including oxidative phosphorylation, glycolysis, glutamine, and polyamine (Fig S3B–E) (Bult et al, 2019). These data reveal a prominent role for Myc in ribosomagenesis and translation more broadly in NK cells, possibly overriding its most established function in directly regulating cell cycle progression.

### Sustained translational capacity is essential for NK cell expansion

We therefore tested the impact of Myc absence on the translation rate of NK cells. To study this process, we assessed the incorporation in newly synthesized proteins of 35S-radiolabeled methionine and cysteine by splenic NK cells exposed for an overnight period to limited or high IL-15 concentration. When acutely stimulated, Myc-deficient NK cells exhibited defective translation (Fig 5A). We therefore verified selected RNA-seq results by Western blot and confirmed that multiple ribosomal proteins were expressed to a lower extent in Myc-deficient NK cells (Fig 5B). This led us to

investigate the effects of low doses cycloheximide, aimed at partially inhibiting translation, on the expansion of WT NK cells. Recapitulating the phenotype of Myc-deficient NK cells, low doses of CHX did not affect NK cell survival at low IL-15, but impaired NK cell expansion in response to high IL-15 concentration (Fig 5C and D). Furthermore, cycloheximide and Myc deficiency limited the raise in NK cell size and granularity observed upon exposure to acute IL-15 stimulation (Fig 5C and E). Taken together, our data underline how a partial defect in translation phenocopies the defects caused by Myc deficiency.

### IL-15 regulates MYC and ribosomal proteins also in human NK cells

In murine NK cells, we observed that IL-15 up-regulated Myc through several mechanisms. We thus aimed to investigate whether this phenomenon was true in human NK cells. For this, we relied on DERL7, a human lymphoma cell line featuring NK cell lineage characteristics (Di Noto et al, 2001; Dufva et al, 2018). Indeed, IL-15 stimulation markedly increased *MYC* transcript and protein levels (Fig 6A and B). Inhibition of the JAK and mTOR pathways and, in particular, concomitant treatment with tofacitinib and Torin2 affected *MYC* mRNA induction (Fig 6A). Similar to mouse NK cells, in DERL7 cells, MYC degradation was accelerated by the blockade of mTOR, whereas ERK pathway inhibition did not affect MYC stability (Fig 6C). These data highlight strong analogies between the regulation of MYC in human and murine NK cells.

To evaluate whether MYC promotes ribosome formation and translation also in human cells, we isolated NK cells from healthy donors' blood and assessed the effects of the specific MYC inhibitor 10058-F4 upon stimulation with 0.25 and 25 ng/ml human IL-15 in vitro. Human NK cells did proliferate significantly already at the low dose of IL-15 and MYC inhibition limited their expansion most prominently in response to high IL-15 stimulation (Figs 6D and E and S4A). Notably, similar effects were achieved by treating human NK cells with low-dose CHX, aimed at partially reducing translation (Figs 6D and E and S4A). Whereas the difference in size and granularity induced on human NK cells going from 0.25 to 25 ng/ml IL-15 was marginal, 10058-F4 and CHX also showed a tendency to reduce NK cell size and granularity (Figs 6F and G and S4A). Notably, 10058-F4 treatment decreased the expression of the tested ribosomal proteins (Fig 6H) induced by—in particular—25 ng/ml of IL-15. These results underline strong parallels between mouse and human NK cells with respect to MYC regulation and its effects on the translation machinery.

We next asked whether ribosomal proteins were affected at the transcriptional level also in primary human NK cells after MYC inhibition. To this aim, we treated primary NK cells with IL-15 in presence or absence of 10058-F4 and checked the mRNA levels of selected ribosomal genes. Transcripts of these genes were however not reduced (Fig S4B). We thus wondered if this could be attributed to the chemical inhibitor used in human NK cells. To assess this, we compared ribosomal gene transcripts in murine control NK cells treated with a Myc inhibitor and in Myc-deficient NK cells. 10058-F4 was not only less effective than Myc ablation to reduce the levels of these transcripts, but also showed transient effects (Fig S4C), indicating that the inhibitor works inefficiently or on some aspects of

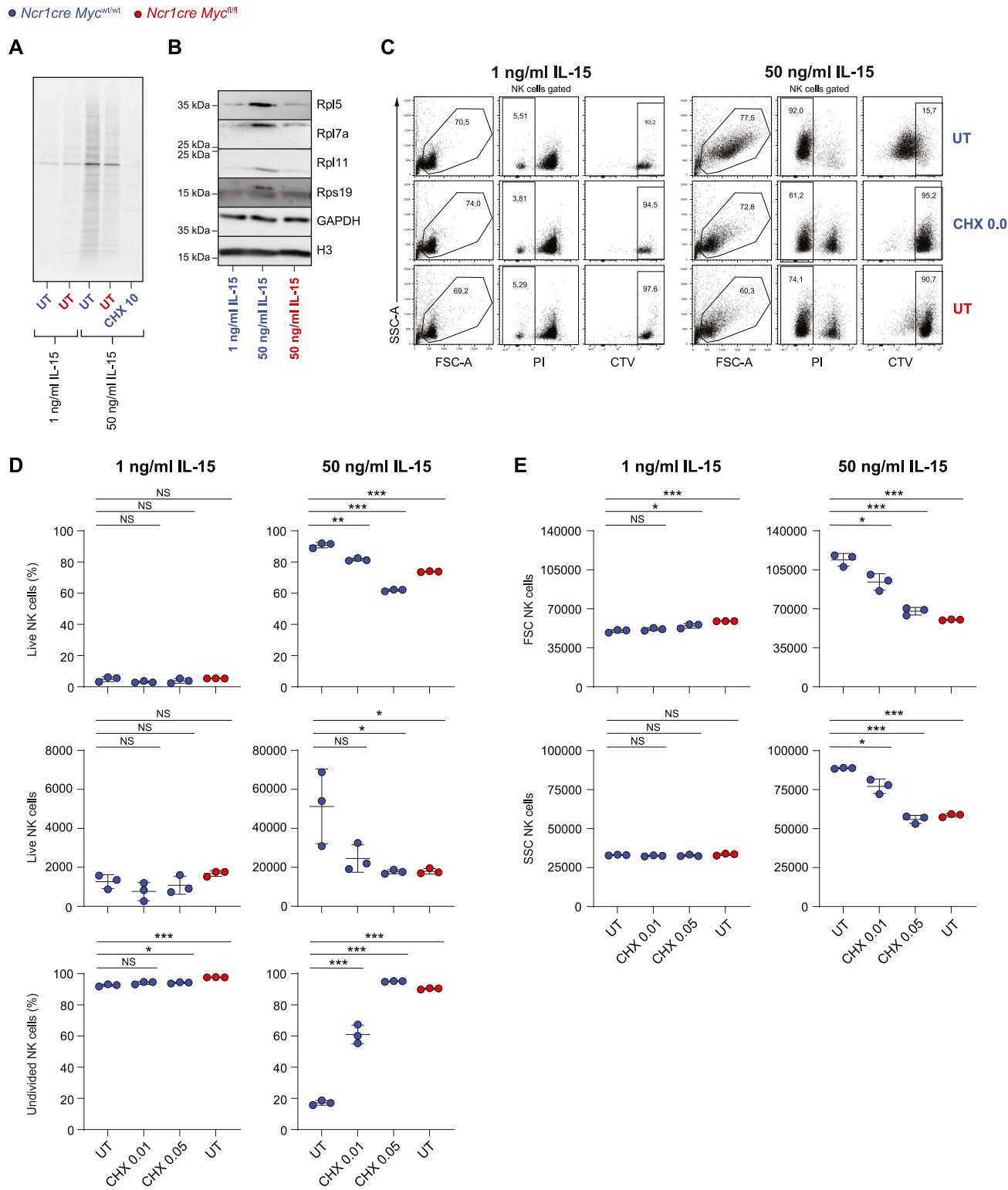

**Figure 5. Full translational capacity is required in response to acute IL-15.**
**(A)** Equal numbers of NK cells enriched from splenocytes of *Ncr1cre Myc*^fl/fl and *Ncr1cre Myc*^wt/wt mice were stimulated overnight with 1 or 50 ng/ml IL-15 and incorporation of 35S-radiolabeled methionine and cysteine was assessed. CHX (10 μg/ml) was added as a control where indicated. **(B)** CD11b⁺ cells (gated as NK1.1⁺CD11b⁺CD3⁻) were sorted from total NK cells enriched from the spleens of *Ncr1cre Myc*^fl/fl and *Ncr1cre Myc*^wt/wt mice and stimulated overnight with IL-15 as indicated. The protein expression levels of Rpl5, Rpl7a, Rpl11, and Rps19 were assessed by Western blotting. Gapdh and Histone H3 levels are shown as loading controls. **(C, D, E)** Enriched Myc-deficient or control splenic NK cells were stimulated with the indicated doses of IL-15. After an overnight period, 0.01 or 0.05 μg/ml of cycloheximide was added and the cells were further cultured for 3 d. **(C)** Representative flow cytometric dot plots gated on NK1.1⁺ cells show live NK cells (PI⁻) or cells division (CTV dilution).

MYC function. In spite of this, MYC inhibition culminated in affecting the translation machinery at the protein level (Fig 6H), suggesting that MYC regulates translation in a robust and multilevel fashion.

### Deletion of Myc impacts on NK cell immunity

We finally asked whether the perturbations caused by the absence of Myc on NK cell homeostasis influenced immunity. Given the importance of inhibitory and activating receptors for NK cell education and function, we first assessed the receptor repertoire on NK cells from *Ncr1cre Myc*$^{fl/fl}$ and *Ncr1cre Myc*$^{wt/wt}$ mice. Although a decrease in the levels of KLRG1 on the effector subsets was noticeable, most other differences were moderate or absent (Fig 7A). A second parameter essential for NK cells' function is their capacity to produce effector cytokines and cytotoxic mediators. As IL-15 is not only a crucial cytokine for NK cell ontogeny and homeostasis, but also for NK cell effector capacity, we measured the production of cytotoxic or inflammatory mediators by Myc-deficient NK cells before or after exposure to IL-15. Although the production of granzyme (Gzm) A was high irrespective of the genotype or the stimulatory condition in DP and CD11b SP NK cells, we observed a substantial defect in the production of GzmB and IFN-γ in DP and CD11b SP Myc-deficient NK cells upon IL-15 stimulation (Fig 7B). These results are in good agreement with the curtailed response to IL-15 that we observed in these NK cell subsets (Fig 3A).

To understand if these differences resulted in a decreased killing capacity by Myc-deficient NK cells, we examined in vitro their cytotoxic activity. Myc-deficient NK cells were able to significantly eliminate RMA-S (lacking MHCI) target cells, despite to a lower extent when compared with control NK cells, a trend that was observed also in the aspecific toxicity affecting control RMA cells (expressing MHCI) (Fig 7C).

Finally, we studied how deletion of Myc affects immunity in vivo. We thus injected *Ncr1cre Myc*$^{fl/fl}$ and *Ncr1cre Myc*$^{wt/wt}$ mice i.v. with B16 cells, a tumor model for metastatic melanoma. Notably, based on the number of nodules found 10 d later in the lung, *Ncr1cre Myc*$^{fl/fl}$ mice eliminated significantly less melanoma cells (Fig 7D). Taken together, these data highlight how *Ncr1cre Myc*$^{fl/fl}$ mice present a substantially weakened antitumor immunity, reflecting a moderate NK cell-intrinsic killing defect together with a substantial loss of mature peripheral NK cells.

## Discussion

Here, we report that Myc is critical for NK cell development and, thus, for NK cell-mediated immunity. With regard to NK cell effector functions, we observed that Myc was needed for the induction of IFN-γ and GzmB by IL-15, in line with findings downstream of IL-2/IL-12 stimulation (Loftus et al, 2018). Indeed, we found that Myc was required for the IL-15-dependent activation of ERK, S6, and STAT5 in effector NK cells. Despite this, neither the levels of granzyme A nor

the killing on a per cell basis were dramatically affected by Myc deletion. Furthermore, with the exception of KLRG1, peripheral Myc-deficient NK cells exhibited mild alterations in the expression of inhibitory receptors. These findings did not support a prime role for Myc in controlling cytotoxicity and inhibitory receptor induction in our system. In *Ncr1cre Myc*$^{fl/fl}$ mice, peripheral mature NK cell numbers were decreased by roughly 70%, largely explaining the defective NK cell-mediated anticancer immunity.

We thus demonstrated a major role for Myc in developing NK cells, in which it is highly expressed. In line with these findings and with previous publications on the role of Myc in IL-15-dependent T cell homeostasis and development (Bianchi et al, 2006; Jiang et al, 2010), we showed that Myc is required for NK cell expansion upon IL-15 stimulation. As IL-15 normally induced multiple signaling pathways in CD27SP Myc-deficient NK cells, we asked whether Myc was implicated downstream of STAT5, mTOR, and ERK. Besides corroborating the role of STAT5 (Gotthardt et al, 2016; Villarino et al, 2017), we identified a contribution by mTOR in regulating *MYC* transcript levels. Furthermore, we found that mTOR is crucial to limit MYC degradation downstream of IL-15. Altogether, these results not only demonstrate that MYC is a central node downstream of IL-15, but also identify the multilevel crosstalk between MYC and IL-15 in murine and human NK cells.

In IL-2/IL-12-stimulated NK cells, Myc has been shown to promote glycolysis and mitochondrial function (Loftus et al, 2018). While we confirm a trend towards decreased expression of genes involved in these pathways in Myc-deficient NK cells, we found that most of the genes significantly regulated by Myc are related to translation, a fundamental cellular process, which might—in turn—broadly affect cellular metabolism. The capacity of Myc to transactivate ribosomal genes has so far been mostly studied in cancer and pluripotent cells (Poortinga et al, 2004; van Riggelen et al, 2010; Seitz et al, 2011; Lee et al, 2012; Scognamiglio et al, 2016; Thomas et al, 2019; Destefanis et al, 2020; Popay et al, 2021). Supporting a role for Myc in basic protein metabolism, a recent study on activated T cells highlighted how Myc controls the expression of amino acid transporters including Slc7a5, whose deletion largely phenocopied the effects of Myc ablation (Marchingo et al, 2020). In NK cells, although we found that Myc maintains *Slc7a5* transcription, Slc7a5 is also required to sustain Myc levels (Loftus et al, 2018), revealing a feed-forward loop concurring to regulate translation. Notably, emerging data show that translation-related genes represent a key but often overlooked signature controlled by Myc in multiple primary immune cells and future work will help define the important roles that such a basic mechanism might play in these cells (Perez-Olivares et al, 2018; Marchingo et al, 2020; Saravia et al, 2020; Tang et al, 2021).

Given the impossibility to genetically rescue the multiple defects of the translational module observed in Myc-deficient NK cells, we explored the effects of partially inhibiting translation in NK cells. Notably, this approach recapitulated the expansion, size, and granularity phenotype observed in both murine and human NK

---

**(D, E)** Graphs show percentages, absolute numbers, percentages of undivided cells (D), FSC, and SSC values (E) of live NK cells. **(A, B, C, D, E)** Results are representative of at least two experiments. **(D, E)** Results show mean ± SD of technical triplicates. **(D, E)** Statistical comparisons are shown; *$P$ ≤ 0.05, **$P$ ≤ 0.01, ***$P$ ≤ 0.001, and NS, non-significant; *t* test, unpaired (D, E).

● Untreated human cells  ● CHX treated human cells  ● MYCi treated human cells

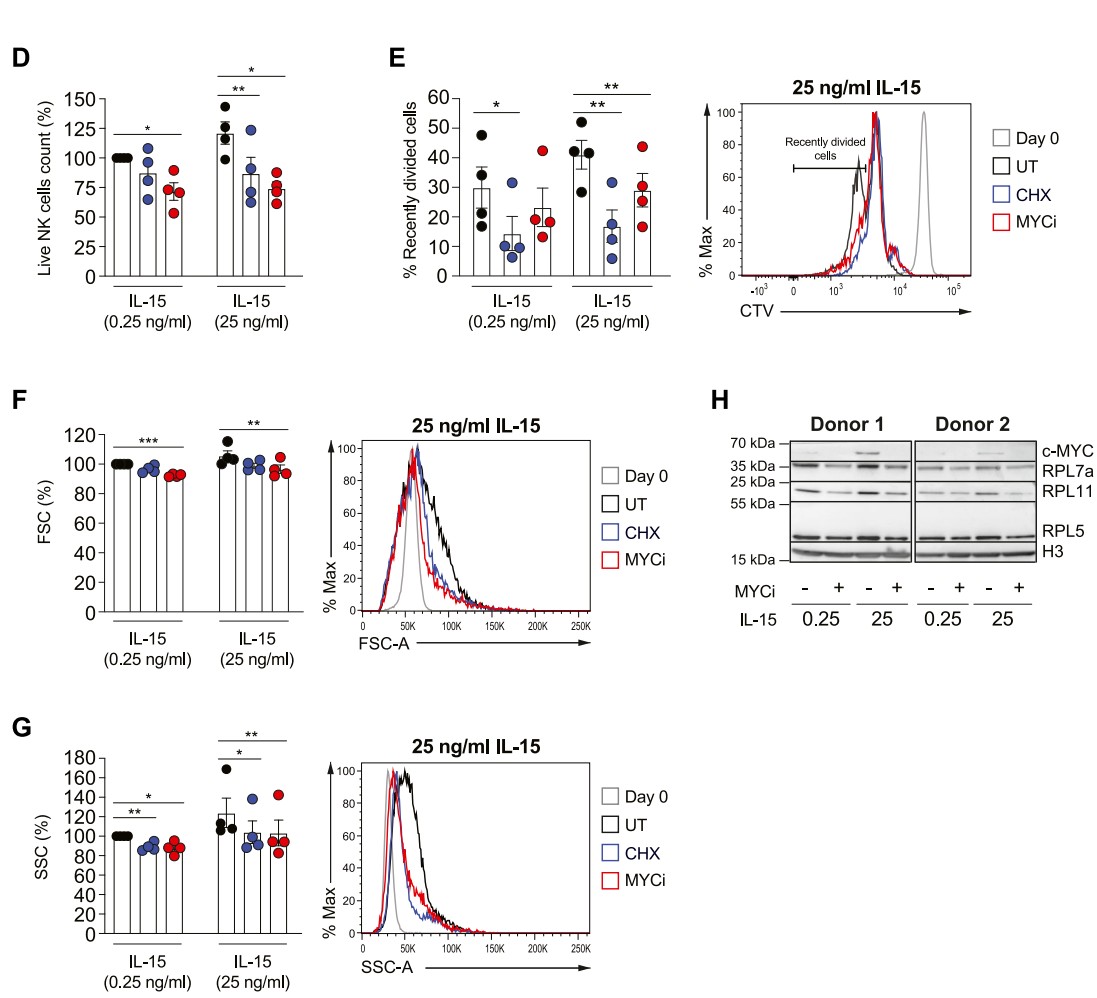

**Figure 6. IL-15 up-regulates MYC and the translational machinery in human NK cells.**
**(A)** qRT–PCR analysis (normalized to *POLR2A*) is shown for *MYC* mRNA in DERL7 cells pretreated (30 min) or not with Torin2 (250 nM), PD98059 (10 $\mu$M), tofacitinib (90 nM) or combinations thereof, and stimulated with 25 ng/ml of IL-15 for 2 h. **(B)** DERL7 cells were stimulated with 25 ng/ml of IL-15 or left untreated for 2 h and MYC expression levels were determined and expressed as geometric MFI and percentage positive; a representative histogram is shown. **(C)** DERL7 cells were stimulated with 25 ng/ml of IL-15 for 2 h and then treated with cycloheximide (20 $\mu$g/ml), Torin2 (250 nM), and/or PD98059 (10 $\mu$M), as indicated. Graphs show the percentage reduction in MYC after 70 min (measured as geometric MFI). **(D, E, F, G)** NK cells enriched from the blood of healthy human donors were stimulated with the indicated doses of IL-15. After an overnight period, 0.04 $\mu$g/ml of CHX or 20 $\mu$M MYC inhibitor 10058-F4 were added on a daily basis and the cells were cultured for further 3 d. NK cells were then analyzed by flow cytometry (CD56$^+$CD3$^-$). **(D, E, F, G)** Graphs show the percentage change in numbers of live NK cells (D), percentage of recently divided, and representative histogram overlay of CTV dilution (E), geometric mean and representative histogram overlays of FSC (F), and SSC (G) in the indicated condition. **(H)** The expressions of MYC and ribosomal proteins RPL5, RPL7a, and RPL11 were assessed in cell lysates by Western blotting; Histone H3 was used as a loading control. **(A, B, C, D, E, F, G)** Results depict

cells lacking functional Myc. However, we also observed that using 10058-F4 to block MYC activity in vitro did not fully mirror the phenotype of Myc-knockout NK cells at the transcriptional level. This suggests that this inhibitor acts either less effectively or differently from genetic deletion. Yet, we conclude that MYC regulates the translational machinery through multiple pathways that remain to be elucidated in greater detail. Importantly, we also identified cycloheximide doses which affected proliferating NK cells but not resting NK cells' viability, revealing an interesting therapeutic window.

Our results suggest that drugs blocking translation, IL-15, JAK or mTOR downstream of this cytokine might concur in the treatment of lymphoid tumors of NK cell origin depending on MYC (Devlin et al, 2016; Schwartz et al, 2017; Laham-Karam et al, 2020). Our data also indicate that the low levels of Myc found in unstimulated peripheral NK cells limit their protein synthesis rate, thereby warranting a status of quiescence, which is important to avoid overzealous responses or a state of terminal differentiation (Castro et al, 2018).

Although the contribution of Ncr1+ ILCs, ILC1s in particular, to the control of B16 melanoma cannot be formally excluded, our observation that liver ILC1s are not reduced and the notion that NK cells are key to control this metastatic cancer model supports the relevance of Myc in conventional NK cells for maintaining the antitumor response (Victorino et al, 2015; Lopez-Soto et al, 2017; Howard et al, 2021). This observation is of great importance as in the work by Tang et al, no significant difference was observed when subcutaneously engrafting B16 melanoma cells in *Ncr1cre Myc*$^{fl/fl}$ mice as compared with the controls (Tang et al, 2021).

Whereas our results suggest that combining NK cell–based immunotherapeutic approaches with strategies aimed at targeting MYC might be inappropriate, data by others suggest that targeting MYC in lymphoma cells exerts beneficial effects on NK cell responses (Swaminathan et al, 2020). Given these complementary observations, it will be important to weight the effects of Myc inhibitors in preclinical cancer models and evaluate different regimens (Han et al, 2019). The multilevel interplay between MYC and IL-15 and the role of this transcription factor in anticancer NK and—most likely—chimeric antigen receptor-NK cells' immunity reinforce the rationale for developing and employing targeted immunostimulatory approaches, such as multifunctional engagers carrying IL-15 (Lee et al, 2011; Vallera et al, 2016; Liu et al, 2018; Gardiner, 2019; Zakiryanova et al, 2019).

# Materials and Methods

### Human primary NK cell isolation and culture

Human peripheral blood mononuclear cells (PBMCs) from healthy donors were prepared form purchased buffy coats (Servizio Trasfusionale della Svizzera italiana; Lugano). Buffy coats were diluted 1:2 by adding an equal amount of prewarmed RPMI 1640 (Gibco) and layered over Ficoll (Cytiva). Cells were span down at 800$g$ for 30 min, at 20°C, without break. Interphase with PBMCs was harvested and diluted with a prewarmed medium followed by spinning down at 450$g$ for 15 min at 20°C and washing with the above medium containing 0.5% human serum. NK cells were enriched from PBMCs by EasySep Human NK Cell Isolation Kit (Stem Cell Technologies) and cultured at 37°C in 5% $CO_2$ in RPMI 1640 (Gibco) supplemented with 10% FBS (Gibco), 50 U/ml penicillin, 50 µg/ml streptomycin (Gibco), 1X MEM NEAA (Gibco), 1 mM sodium pyruvate (Gibco) in the presence of the indicated amounts of human recombinant IL-15 (Peprotech).

### Human NK cell lines' culture

DERL7 cell line was kindly provided by Prof. F Bertoni, IOR, Bellinzona, and cultured at 37°C in 5% CO2 in RPMI 1640 (Gibco) supplemented with 20% FBS (Gibco), 50 U/ml penicillin, 50 µg/ml streptomycin (Gibco), 1 mM sodium pyruvate (Gibco) in the presence of 100 IU/ml recombinant human IL-2 (PeproTech).

### Mice

*Ncr1cre Myc*$^{fl/fl}$ and *Ncr1cre Myc*$^{wt/wt}$ mice were generated by crossing *Ncr1cre* and *Myc*$^{fl/fl}$ mice which were previously described (Trumpp et al, 2001; Narni-Mancinelli et al 2011). *Ncr1cre Myc*$^{wt/wt}$ mice were also crossed onto a CD45.1 congenic background. The mice were backcrossed multiple times on a C57BL/6 background. These mice and C57BL/6 mice were crossed and/or housed at the animal facility of the University of Lausanne and of the Institute for Research in Biomedicine of Bellinzona. Sex- and age-matched 6–14-wk-old mice were used in the experiments. All animal experimental protocols were approved by the veterinary office regulations of the states of Vaud and of Ticino, Switzerland, and all methods were performed in accordance with the Swiss guidelines and regulations.

### Isolation of liver immune cells

Livers were perfused with PBS 10 IU/ml heparin (Sigma-Aldrich) and mechanically dissociated by gently pressing through 40 µM nylon mesh filters. The cell suspension was centrifuged at 50$g$ for 5 min to remove the parenchymal cells. The supernatants were centrifuged for 5 min at 500$g$ and the pellet was washed twice with RPMI 1640 (Gibco) medium. The pellet was then resuspended in 40% Percoll (Cytiva) in HBSS (Gibco) and gently overlaid onto 70% Percoll and centrifuged at 800$g$ for 25 min without a break. The cells were then collected at the interface and washed with PBS and used for flowcytometry analysis.

mean ± SEM of n = 3 biological replicates and are a pool of three independent experiments (A), mean ± SD of n = 3 technical replicates and are representative of two independent experiments (B, C), and mean ± SEM of n = 4 donors (D, E, F, G; the average of the condition treated with 0.25 ng/ml IL-15 only was set as 100%). **(A, B, C, D, E, F, G)** Statistical comparisons are shown; *$P \leq 0.05$, **$P \leq 0.01$, and ***$P \leq 0.001$; $t$ test, paired (A, D, E, F, G) or unpaired (B, C); only statistically significant differences are shown.

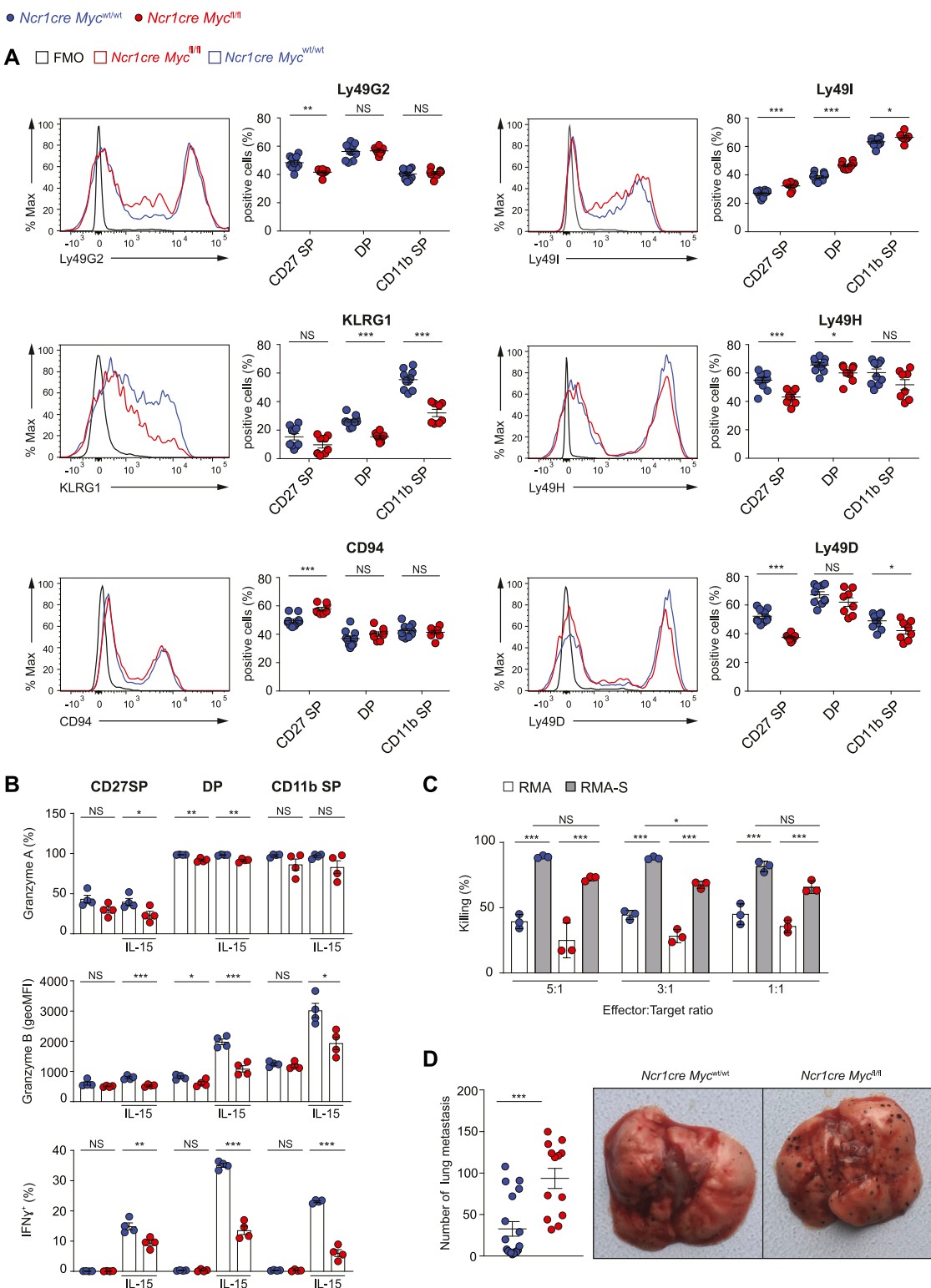

**Figure 7. NK cell-intrinsic requirement of Myc for B16 control.**
**(A)** The expression of the indicated receptors on splenic NK cells (NK1.1⁺CD3⁻CD19⁻) from *Ncr1cre Myc*^fl/fl and *Ncr1cre Myc*^wt/wt mice is depicted as representative flow cytometry histogram and as percentage positive population. **(B)** Graph illustrates the production of granzyme A, granzyme B, and IFN-γ by splenic NK cells (gated as NK1.1⁺CD3/CD19⁻) after stimulation with 50 ng/ml IL-15 for 4.5 h as measured by flow cytometry. **(C)** NK cells isolated from polyinosinic:polycytidylic acid (polyI:C)-treated *Ncr1cre Myc*^fl/fl and *Ncr1cre Myc*^wt/wt mice were plated with RMA or RMA-S cells at the indicated ratios. The graph depicts percentage killing of target cells, as measured by quantifying PI⁻ living target cells after 4 h. **(D)** Quantification of nodules and a representative picture of the lungs at day 10 after intravenous injection of B16F10 melanoma. **(A, B, D)** Results illustrate mean ± SEM of n = 8–10 mice per genotype (A), of n = 4 mice per genotype (B), of n = 13–18 mice per genotype (D), and are a pool of two

## Melanoma tumor model

B16-F10 melanoma cells were maintained in RPMI-1640 (Gibco) supplemented with 10% FBS (Gibco), 2 mM L-glutamine (Gibco), 50 U/ml penicillin, 50 μg/ml streptomycin (Gibco) and cultured at 37°C in 5% $CO_2$. Then, $10^5$ cells were intravenously injected in *Ncr1cre Myc*$^{wt/wt}$ mice and *Ncr1cre Myc*$^{fl/fl}$ mice. The mice were euthanized at day 10 after injection and the number of lung surface nodules was counted.

## Mixed BM chimeras

Mixed BM chimeras were generated as previously described (Niogret et al, 2019), except that donor BM cells were obtained from sex-matched *Ncr1cre Myc*$^{fl/fl}$ mice and *Ncr1cre Myc*$^{wt/wt}$ mice with the CD45.1 congenic marker by flushing tibias and femurs and mixing them in a 1:1 ratio before injection. The chimeras were used for the respective experiments 6–8 wk post-BM reconstitution.

## Flow cytometry

For flow cytometry analysis, murine cells were preincubated with anti-CD16/32 (2.4G2) to block Fc receptors and then surface stained for 20–30 min at 4°C with antibodies to CD3e (145-2C11), CD11b (M1/70), CD19 (1D3 or 6D5), CD27 (LG.7F9 or LG.3A10), CD45.1 (A20), CD45.2 (104), CD94 (18D3), CD122 (TM-b1), CD132 (TUGm2), CD49a (HMα1), DX5/CD49b (DX5), Ncr1 (29A1.4), NK1.1 (PK136), KLRG1 (2F1), Ly49D (4 × $10^5$), Ly49G2 (4D11), Ly49I (YLI-90), and Ly49H (3D10) (purchased from BioLegend or eBioscience).

Human cells were surface stained for 25 min at 4°C with antibodies anti-CD3 (Clone UCHT1; eBioscience) and anti-CD56 (Clone 5.1H11; BioLegend).

Streptavidin-conjugated fluorophores were purchased from BioLegend or eBioscience. Stainings were performed with appropriate combinations of fluorophores. Propidium iodide (0.05 μg/ml from Immunochemistry) was used as a live–dead marker. Events were collected on a FACS Canto, Fortessa (Becton Dickinson) and data were analyzed using FlowJo software (Tree Star Inc.).

## Intracellular flow cytometry stainings

For intracellular staining of Ki67 (SolA15, 1:100; eBioscience), murine NK cells were first surface stained, fixed, and permeabilized using the FoxP3 transcription factor staining buffer set of eBioscience (00-5523-00) according to the manufacturer's recommendations. For assessment of Myc levels, intracellular staining was performed using Foxp3/Transcription Factor Staining Buffer Set (eBioscience) according to the manufacturer's protocol. Cells were stained for 1 h in a permeabilization buffer with an Myc antibody (Myc/n-Myc D3N8F rabbit mAb PE antibody, #35876, 1:50; Cell Signaling). Myc expression (expressed as geometric MFI) was measured by flow cytometry and the background of Myc-knockout NK cells was

subtracted. For Bcl2 staining, BM cells were fixed, permeabilized as above, and stained for 1 h in the permeabilization buffer with Bcl2 antibody (Bcl2 10C4, 1:100; Invitrogen).

To assess the Myc stability, enriched mouse splenic NK cells or human DERL7 NK cell line were stimulated 2 or 3 h, respectively, with recombinant mouse or human IL-15. The cells were then treated with CHX (20 μg/ml) alone or in combination with Torin2 (250 nM from Selleckchem), PD98059 (10 μM) or both for 70 min. The Myc content was then assessed by intracellular staining as mentioned above.

## Analyses of effector molecule production

For intracellular IFN-γ, GzmA, and GzmB detection, splenocytes from control and *Ncr1cre Myc*$^{fl/fl}$ mice were left unstimulated or stimulated for a total of 4.5 h at 37°C with 50 ng/ml IL-15 and Brefeldin A (10 μg/ml Enzo Life Science) was added after the first 2 h. The cells were fixed with fixation/permeabilization buffer (eBioscience) for intracellular staining for IFN-γ (F3 IGH48; eBioscience) and granzyme A (3G8.5; eBioscience). Granzyme B (NGZB; eBioscience) staining was performed after fixation (25 min at 22–28°C) and permeabilization with the Foxp3 Transcription Factor Staining Buffer Set (#00-5523-00; eBioscience). Intracellular staining was then performed for 30 min at 22–28°C (with the antibody diluted in the permeabilization buffer).

## Signaling analysis

For signaling analysis, splenocytes from control and *Ncr1cre Myc*$^{fl/fl}$ mice were left unstimulated or stimulated with 50 ng/ml recombinant mouse IL-15 (PeproTech) for 40 min at 37°C and analyzed for intracellular protein phosphorylation by flow cytometry. The cells were then surface stained on ice, fixed with 2% paraformaldehyde at 4°C for 20 min, and permeabilized with 90% methanol for 30 min at 4°C. Intracellular phosphostainings were performed 1 h at room temperature in the dark with antibodies against phospho-STAT5 (Tyr694; D47E7 XP rabbit mAb #4322; Cell Signaling, 1:150), phospho-S6 ribosomal protein (Ser235/236; D57.2.2E; XP rabbit mAb #4858; Cell Signaling, 1:200), phospho-ERK1/2 (Thr202/Tyr204; D13.14.4E; XP rabbit mAb #4370; Cell Signaling, 1:150).

## Mouse NK cell isolation and culture

For experiments requiring enriched NK cells, splenic NK cells were isolated using the EasySep mouse NK cell isolation kit (Stemcell Technologies) according to the manufacturer's recommendations. NK cell enrichment was confirmed by flow cytometry. As indicated, NK cells were further FACS-sorted for selected experimental settings and when derived from BM. NK cells were then cultured in RPMI 1640 (Gibco) supplemented with 10% FBS (Gibco), 50 U/ml penicillin, 50 μg/ml streptomycin (Gibco), 1 mM sodium pyruvate (Gibco), 50 μM 2-mercaptoethanol (Gibco), and cultured at 37°C in

experiments (A, D) or are representative of at least two experiments (B). **(A, B, D)** Each symbol represents an individual mouse. **(C)** Results depict mean ± SD (n = 3 technical replicates) and are representative of at least two experiments. **(A, B, C, D)** Statistical comparisons are shown; *P ≤ 0.05, **P ≤ 0.01, ***P ≤ 0.001, and NS, non-significant; *t* test, unpaired (A, B, D) and two-way ANOVA (C).

5% $CO_2$ with the indicated amounts of recombinant mouse IL-15 (PeproTech).

## Immunoblot analyses

Human or mouse NK cells or NK cell lines were lysed in Laemmli sample buffer, boiled for 5 min at 95°C, and separated on SDS–PAGE gel. The proteins were then transferred onto a nitrocellulose membrane, blocked by 5% milk in PBS TWEEN 20 (Sigma-Aldrich), and then incubated with the primary antibodies: anti-Myc (D84C12), anti-RPL5 (D5Q5X), anti-RPL7a (E109), anti-RPL11(D1P5N), anti-Gapdh (polyclonal), anti-β-Actin (8H10D10), anti-S6 ribosomal protein (54D2), anti-p42 MAP kinase (Erk2), anti-Stat5 (D3N2B), anti-P-p44/42 MAPK (T202/Y204) (D13.14.4E), anti-P-S6 (S235/236) (D57.2.2E), anti-P-p70 S6 kinase (T389) (108D2), anti-P-Stat5 (Y649) (D47E7), anti-Histone H3 (D1H2) from Cells Signaling Technologies (1:1,000 dilution) and anti-RPS19 (ab40833; Abcam, 0.2 µg/ml). Blots were washed and incubated with respective secondary antibodies (1:2,500) followed by washing and developed using a chemiluminescent substrate solution (Witec AG) and imaged by image analyzer machine Fusion FX (Vilber).

## Metabolic labeling

Mouse NK cells (purity > 94%) were washed with prewarmed (PBS; Gibco) and then pulse labeled with 0.05 mCi of [35S]-methionine/cysteine in DMEM (Gibco), 50 mM Hepes (Gibco), and 2 mM L-glutamine (Gibco) for 20 min at 37°C. The labeling medium was removed and the cells were washed with PBS. The cells were lysed with RIPA buffer (1% Triton X-100, 0.1% SDS, 0.5% sodium deoxycholate in HBS, pH 7.4) containing 20 mM N-Ethylmaleimide and protease inhibitor cocktail (1 mM PMSF, chymostatin, leupeptin, antipain, and pepstatin, 10 µg/ml each) for 20 min on ice. Lysates were subjected to centrifugation at 4°C and 10,000$g$ for 10 min and postnuclear supernatants were collected. Samples were subjected to SDS–PAGE, and proteins were transferred to PVDF membranes using the Trans-Blot Turbo Transfer System (Bio-Rad). PVDF membranes were exposed to autoradiography films (GEHealthcare) and were scanned with the Typhoon FLA 9500 (Software Version 1.0).

## Cell labeling and in vitro expansion analyses

To study murine NK cell expansion, enriched NK cells were labeled with 5 µM cell trace violet (CTV; Life Technologies) in PBS 2% FBS at 37°C for 20 min. Cells were seeded at a density of 0.6–2 × 10^6/ml in the presence of the indicated concentration of recombinant mouse IL-15. In the experiments where cycloheximide (CHX; Sigma-Aldrich) was employed, after an overnight period, 0.01 or 0.05 µg/ml of CHX was added and the cells were further cultured for 3 d.

To study human NK cell expansion, enriched NK cells were labeled with 5 µM CTV, seeded at a density of 0.5 × 10^6/ml and stimulated with 0.5 or 25 ng/ml recombinant human IL-15 (PeproTech). After an overnight period, 0.04 µg/ml of CHX, 20 µM MYC inhibitor 10058-F4 (Sigma-Aldrich) was added, and the cells further cultured for 3 d. The cells were stained and analyzed by flow cytometry.

## In vitro killing assay

RMA and RMA-S cell lines were cultured in RPMI 1640 (Gibco) supplemented with 10% FBS (Gibco), 50 U/ml penicillin, 50 µg/ml streptomycin (Gibco), 1 mM sodium pyruvate (Gibco), 50 µM 2-mercaptoethanol (Gibco), at 37°C with 5% $CO_2$. Control *Ncr1cre Myc*^wt/wt and *Ncr1cre Myc*^fl/fl recipient mice were pre-treated with 150 µg of polyinosinic:polycytidylic acid (Poly[I:C]; InvivoGen) by intraperitoneal injection 24 h before the experiment. Splenocytes were collected and enriched for NK cells (Stemcell Technologies). 9 × 10^3 CTV-labeled target cells (RMA or RMA-S) were then cocultured in NK cell medium for 4 h at 37°C in a 1:1, 1:3 or 1:5 ratio to the NK effector cells in a 96-well plate. Analysis was then performed by flow cytometry. RMA rejection is shown as percentage of killing.

## Quantitative RT–PCR analyses

BM and splenic NK cells were FACS-sorted as indicated in the text. Sorts were performed using FACSAria III, BD Biosciences. In other experiments, isolated WT NK cells were pretreated (15–30 min) or not with Torin2 (250 nM from Selleckchem), PD98059 (10 µM from Adipogen), tofacitinib (90 nM from GSK) or combinations thereof and stimulated with 50 ng/ml of recombinant mouse IL-15 for 2 h. Then, total RNA was extracted using the TRIzol reagent (Life Technologies) according to the manufacturer's instructions. Annealing with random primers (Life technologies) was performed at 70°C for 5 min, followed by retrotranscription to cDNA with M-MLV RT, RNase H(−) point mutant (Promega) in the presence of buffer, nucleotides, and RNasin Plus RNase Inhibitor (Promega). The reaction was then incubated at 40°C for 10 min, 45°C for 50 min, and 70°C for 15 min. cDNAs obtained were diluted and quantitative PCR was performed using the PerfeCTa SYBR Green FastMix (Quantabio) on a QuantStudio 3 Real-Time PCR System machine (Applied Biosystems). Standard cycling was used (45 cycles of 95, 60, and 72°C steps of 10 s each). Negative control reactions were cycled alongside test samples to ensure the absence of contaminating genomic DNA. The expression of *Myc* gene was determined relative to the abundance of the housekeeping gene RNA polymerase II Subunit A (*Polr2a*) and/or 18 s. Data were analyzed and transcript abundance and SD calculated using the Thermo Fisher Connect software.

Primers (Fwd; Rev) used were as follows; for mouse: *Polr2a*: (5′-CCGGATGAATTGAAGCGGATGT-3′; 5′-CCTGCCGTGGATCCATTAGTCC-3′); *Myc*: (5′-CCAGCAGCGACTCTGAAGAA-3′; 5′-GGCAGGGGTTTGCCTCTT-3′); *18 s*: (5′-GTAACCCGTTGAACCCCATT-3′; 5′-CCATCCAATCGGTAGTAGCG-3′), *Rplp0* (5′-AGATTCGGGATATGCTGTTGGC-3′; 5′-TCGGGTCCTAGACCAGTGTTC-3′), *Rpl5*: (5′-TTGGTGATCCAGGACAAGAATAA-3′; 5′-GCACAGACGATCATATCCCC-3′), *Rpl12*: (5′-ATCAAAGCCCTCAAGGAGCC-3′; 5′-AAGACCGGTGTCTCATCTGC-3′), *Rps19*: (5′-CAGCACGGCACCTGTACCT-3′; 5′-GCTGGGTCTGACACCGTTTC-3′), *Gapdh*: (5′-TGATGGGTGTGAACCACGAG-3′; 5′-GCCCTTCCACAATGCCAAAG-3′), *Hprt*: (5′-GCAGTACAGCCCCAAAATGG-3′; 5′-AACAAAGTCTGGCCTGTATCCAA-3′).

For human: *Polr2a*: (5′-CGCACCATCAAGAGAGTCCAGTTC-3′; 5′-GTATTTGATGCCACCCTCCGTCA-3′); *Myc*: (5′-GCTGCTTAGACGCTGGATT-3′; 5′-CGAGGTCATAGTTCCTGTTGG-3′), *Ube2d2*: (5′-GATCACAGTGGTCTCCAGCA-3′; 5′-CGAGCAATCTCAGGCACTAA-3′), *Hprt*: (5′-TCAGGCAGTATAATCCAAAGATGGT-3′; 5′-AGTCTGGCTTATATCCAACACTTCG-3′), *Rpl7a*: (5′-GCTGA

AAGTGCCTCCTGCGA-3′; 5′-CACCAAGGTGGTGACGGTGT-3′), *Rpl11*: (5′-TCCAC TGCACAGTTCGAGGG-3′; 5′-AAACCTGGCCTACCCAGCAC-3′), and *Rpl14*: (5′-TTAAGAAGCTTCAAAAGGCAGC-3′; 5′-TTTTGACCCTTCTGAGCTTTTG-3′).

### Library preparation and sequencing

Total RNA was extracted from sorted cells with an RNeasy plus mini kit (QIAGEN) according to the manufacturer's instructions. The quantity and quality of the isolated RNA was determined with a Qubit (1.0) Fluorometer (Life Technologies) and a 4200 TapeStation System (Agilent). The libraries were prepared following the Takara SMART-Seq v4 Ultra Low Input RNA Kit, with PolyA selection. This protocol relies on the template switching activity of reverse transcriptases to enrich for full-length cDNAs and to add defined PCR adapters directly to both ends of the first-strand cDNA. Briefly, total RNA samples (0.25–10 ng) were reverse transcribed using random priming into double-stranded cDNA in the presence of a template switch oligo. When the SMARTScribe Reverse Transcriptase reaches the 5′ end of the RNA, its terminal transferase activity adds a few nucleotides to the 3′ end of the cDNA. The SMART-Seq Oligonucleotide base-pairs with the non-templated nucleotide stretch, creating an extended template to allow the reverse transcriptase to continue replicating to the end of the oligonucleotide. The SMART-Seq primer and oligo serve as universal priming sites for cDNA amplification by PCR. 0.5 ng of cDNA from each sample were tagmented and amplified using Illumina Nextera XT kit. The resulting libraries were pooled, double-sided size selected (0.5× followed by 0.8× ratio using Beckman Ampure XP beads), and quantified using an Agilent 4200 TapeStation System.

For the CD27SP NK cell isolated from the BM, after quantification, libraries were prepared for loading accordingly to the NovaSeq workflow with the NovaSeq6000 Reagent Kit (Illumina). Cluster generation and sequencing were performed on a NovaSeq6000 System with a run configuration of single end 100 bp with a depth of around 20 Mio reads per sample. FASTQ files were processed using Nextflow and the nfcore/RNAseq pipeline with default values (Di Tommaso et al, 2017; Ewels et al, 2020). Quantification of transcript expression on the gene level was computed using Salmon and the GRCh38 reference transcriptome (Patro et al, 2017).

### RNA-seq analyses

For gene-level testing, statistical significance was assessed using three independent statistical methods to be more robust: DESeq2 (Wald test), edgeR (QLF test), and limma-trend (Robinson et al, 2010; Love et al, 2014; Ritchie et al, 2015). The maximum q-value of the three methods was taken as aggregate q-value, which corresponds to taking the intersection of significant genes from all three tests. Statistical testing and visualization (heatmap, volcano plot) have been performed using the Omics Playground version v2.7.8 (Akhmedov et al, 2020). Gene set enrichment analysis was performed using the KEGG gene sets (Kanehisa & Goto, 2000) and a combination of three statistical tests: fGSEA (Korotkevich, 2021 *Preprint*), (Fisher, 1922), and GSVA (Hanzelmann et al, 2013).

### Statistical analyses

Unless otherwise described, unpaired *t* test were performed using Prism software (version 8.2.0; GraphPad). Where indicated, paired *t* test or two-way ANOVA test were performed using Prism software (version 8.2.0; GraphPad). Differences were considered significant when $P \leq 0.05$ (*), very significant when $P \leq 0.01$ (**), and highly significant when $P \leq 0.001$ (***).

## Data Availability

RNA-seq datasets are available in the GEO repository under the number GSE220930. The data that support the findings of this study are available from the corresponding author upon reasonable request.

## Supplementary Information

## Acknowledgements

We thank S Chelbi, M Akhmedov, D Jarrossay, T Soldà, M Molinari, and the Animal Facility collaborators at the IRB, Bellinzona, and P Paganetti, EOC, Bellinzona, for precious help and/or advice. We thank W Held, Lausanne, for sharing reagents and/or technical help. We thank B Fischer, IRB Bellinzona, for critical reading of the manuscript. Funding: This work was supported by the Leonardo Foundation, Lugano, the Novartis Foundation, Basel, the Fondazione San Salvatore, Lugano, the Swiss National Science Foundation (310030_185185 and 310030_197771), the Swiss Cancer Research Foundation (KFS 5141-08-2020), and the European Research Council (ERC-2012-StG310890) to G Guarda. This work was partly supported by the ERC advanced Grant SHATTER-AML (ERC-2022-AdG101055270) and the Dietmar Hopp Foundation (to A Trumpp).

### Author Contributions

HJ Khameneh: conceptualization, data curation, formal analysis, investigation, and writing—original draft, review, and editing.
N Fonta: conceptualization, data curation, formal analysis, and investigation.
A Zenobi: conceptualization, data curation, formal analysis, investigation, and writing—original draft, review, and editing.
C Niogret: conceptualization, data curation, formal analysis, and investigation.
P Ventura: investigation.
C Guerra: investigation.
I Kwee: formal analysis and methodology.
A Rinaldi: methodology.
M Pecoraro: methodology.
R Geiger: methodology.
A Cavalli: methodology.
F Bertoni: resources.
E Vivier: resources.

A Trumpp: resources.
G Guarda: conceptualization, data curation, supervision, funding acquisition, project administration, and writing—original draft, review, and editing.

## Conflict of Interest Statement

Unrelated projects in G Guarda laboratory are supported by OM-Pharma, Meyrin, and IFM Therapeutics, Boston. F Bertoni has research funds from Acerta, ADC Therapeutics, Basilea Pharmaceutica, Bayer AG, Cellestia, CTI Life Sciences, Helsinn, ImmunoGen, Menarini Ricerche, NEOMED Therapeutics 1, Nordic Nanovector, Oncology Therapeutic Development, Oncternal Therapeutics, PIQUR Therapeutics, Spexis; consultancy fee from Helsinn, Menarini; expert statements provided to HTG; travel grants from Amgen, Astra Zeneca, Jazz Pharmaceuticals, PIQUR Therapeutics AG. R Geiger is a compensated cofounder and member of the scientific executive board of Encentrio Therapeutics. R Geiger owns stock in Encentrio Therapeutics. E Vivier is an employee of Innate-Pharma, Marseille.

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
