## [Reviewer comments · Life Science Alliance]

Myc controls NK cell development, IL-15-driven expansion, and translational machinery

Hanif J. Khameneh, Nicolas Fonta, Alessandro Zenobi, Charlène Niogret, Pedro Ventura, Concetta Guerra, Ivo Kwee, Andrea Rinaldi, Matteo Pecoraro, Roger Geiger, Andrea Cavalli, Francesco Bertoni, Eric Vivier, Andreas Trumpp and Greta Guarda

DOI: 10.26508/lsa.202302069

Corresponding author(s): Prof. Greta Guarda (Università della Svizzera italiana)

Review timeline:

Submission Date:	2023-04-02
Editorial Decision:	2023-04-03
Revision Received:	2023-04-05
Editorial Decision:	2023-04-11
Revision Received:	2023-04-19
Accepted:	2023-04-19

Scientific Editor: Eric Sawey

Transaction Report:

Please note that the manuscript was previously reviewed at another journal and the reports were taken into account in the decision-making process at Life Science Alliance.

No Peer Review Process File is available with this article, as the authors have chosen not to make the review process public in this case.

Re: Life Science Alliance manuscript #LSA-2023-02069-T

Prof. Greta Guarda
Institute for Research in Biomedicine
Via Chiesa 5
Bellinzona, Ticino 6500
Switzerland

Dear Dr. Guarda,

Thank you for submitting your manuscript entitled "Myc controls IL-15-driven expansion and translational machinery of NK cells" to Life Science Alliance. We invite you to submit a revised manuscript addressing the following Reviewer comments:

- Address Reviewer 2's comments.
- Address Reviewer 3's minor concerns.
- Place this work into context of earlier work, including PMID 34193438.

Thank you for this interesting contribution to Life Science Alliance. We are looking forward to receiving your revised manuscript.

Sincerely,

B. MANUSCRIPT ORGANIZATION AND FORMATTING:

RE: Life Science Alliance Manuscript #LSA-2023-02069-TR

Prof. Greta Guarda
Università della Svizzera italiana
Institute for Research in Biomedicine
Via Chiesa 5
Bellinzona, Ticino 6500
Switzerland

Dear Dr. Guarda,

Thank you for submitting your revised manuscript entitled "Myc controls NK cell development, IL-15-driven expansion, and translational machinery". We would be happy to publish your paper in Life Science Alliance pending final revisions necessary to meet our formatting guidelines.

- please consult our manuscript preparation guidelines <https://www.life-science-alliance.org/manuscript-prep> and make sure your manuscript sections are in the correct order
- please add a conflict of interest statement to your main manuscript text
- label EV figures as Supplemental Figures and adjust the naming and related callouts (S1a, etc) within the text to reflect this update
- please update the Data Availability statement to include the GEO accession info for the RNA-seq datasets, and these should be made publicly accessible at this stage

Figure Check:

- please add sizes next to all blots

A. FINAL FILES:

B. MANUSCRIPT ORGANIZATION AND FORMATTING:

Sincerely,

3rd Editorial Decision

19 April 2023

RE: Life Science Alliance Manuscript #LSA-2023-02069-TRR

Prof. Greta Guarda
Università della Svizzera italiana
Institute for Research in Biomedicine
Via Chiesa 5
Bellinzona, Ticino 6500
Switzerland

Dear Dr. Guarda,

Thank you for submitting your Research Article entitled "Myc controls NK cell development, IL-15-driven expansion, and translational machinery". It is a pleasure to let you know that your manuscript is now accepted for publication in Life Science Alliance. Congratulations on this interesting work.

DISTRIBUTION OF MATERIALS:

Again, congratulations on a very nice paper. I hope you found the review process to be constructive and are pleased with how the manuscript was handled editorially. We look forward to future exciting submissions from your lab.

Sincerely,

Eric Sawey, PhD
Executive Editor

Life Science Alliance
<http://www.lsajournal.org>